# Process Analysis and Topography Evaluation for Monocrystalline Silicon Laser Cutting-Off

**DOI:** 10.3390/mi14081542

**Published:** 2023-07-31

**Authors:** Fei Liu, Aiwu Yu, Chongjun Wu, Steven Y. Liang

**Affiliations:** 1College of Mechanical Engineering, Donghua University, Shanghai 201620, China; lf2871157603@163.com; 2Shanghai Aircraft Manufacturing Co., Ltd., Shanghai 201620, China; calebyu1988@163.com; 3Manufacturing Research Center, Georgia Institute of Technology, Atlanta, GA 30332, USA; steven.liang@me.gatech.edu

**Keywords:** monocrystalline silicon, laser cutting, ablation zone, quality analysis

## Abstract

Due to the characteristics of high brittleness and low fracture toughness of monocrystalline silicon, its high precision and high-quality cutting have great challenges. Aiming at the urgent need of wafer cutting with high efficiency, this paper investigates the influence law of different laser processes on the size of the groove and the machining affected zone of laser cutting. The experimental results show that when laser cutting monocrystalline silicon, in addition to generating a groove, there will also be a machining affected zone on both sides of the groove and the size of both will directly affect the cutting quality. After wiping the thermal products generated by cutting on the material surface, the machining affected zone and the recast layer in the cutting seam can basically be eliminated to generate a wider cutting seam and the surface after wiping is basically the same as that before cutting. Increasing the laser cutting times will increase the width of the material’s machining affected zone and the groove width after chip removal. When the cutting times are less than 80, increasing the cutting times will increase the groove width at the same time; but, after the cutting times exceed 80, the groove width abruptly decreases and then slowly increases. In addition, the lower the laser scanning speed, the larger the width of the material’s machining affected zone and the width of the groove after chip removal. The increase in laser frequency will increase the crack width and the crack width after chip removal but decrease the machining affected zone width. The laser pulse width has a certain effect on the cutting quality but it does not show regularity. When the pulse width is 0.3 ns the cutting quality is the best and when the pulse width is 0.15 ns the cutting quality is the worst.

## 1. Introduction

In recent years, semiconductor technologies have developed rapidly. Among them, monocrystalline silicon is a typical orthotropic hard brittle semiconductor material [1,2,3] which has excellent thermal conductivity, mechanical strength, overload resistance, and other properties [4,5]. It has been widely used in aerospace, national defense construction, biotechnology, and other aspects [6,7,8]. However, the high brittleness and low fracture toughness of monocrystalline silicon make it a typically difficult to machine material [5,9,10].

The initial method for cutting crystalline silicon is mortar wire cutting but this method has a low cutting efficiency, high material loss, and a certain degree of water pollution so it is gradually replaced by diamond wire cutting. The diamond wire-cutting method has a high cutting efficiency and low silicon material loss [11] so it has become one of the key technologies in the solar cell manufacturing process and semiconductor chip manufacturing process [12,13]. Costa et al. [14] discussed the influence of diamond wire cutting on the feed force and the quality of monocrystalline silicon wafers, found a method to reduce the feed force, and obtained smoother material surface and shallower microcracks. Costa et al. [15] investigated the effect of diamond wire saws on the surface integrity of monocrystalline silicon and used circular diamond wire saws to cut the monocrystalline silicon. They found that the most suitable cutting parameters were the lowest feed rate and wire tension and the highest wire cutting speed. Wang et al. [16] investigated the effect of the scribing speed on the surface morphology and material removal behavior of monocrystalline silicon diamond wire saws by designing high-speed diamond scribing experiments and explained the potential mechanism of promoting brittle fracture at higher scratching speeds. But the cutting efficiency of traditional diamond wire cutting is still not high; it is far from meeting the market demand.

With the development of science and technology, laser cutting technology is becoming more and more mature [17]. Laser cutting technology is one of the most widely used non-contact material cutting methods [18,19] which has the advantages of a narrow groove, small machining affected zone, fast cutting speed, higher precision, and good controllability [20,21]. Zhao et al. [22] used a laser to cut anisotropic monocrystalline silicon bilayer wafers and found that the anisotropy of monocrystalline silicon had a great impact on the quality of crack edge morphology and the form of silicon fracture after cutting. Mulugeta et al. [23] used laser technology to cut silicon anodes and determined the minimum average power and energy efficiency of four cutting widths and five cutting methods.

However, due to the immature laser cutting technology, the processing effect of monocrystalline silicon is not ideal [24,25], especially since the machining affected zone cannot be completely eliminated in the laser direct cutting [26] and the excessive machining affected zone will reduce the chip performance. Huang et al. [27] used picosecond lasers with different fluences to cut ultra-thin wafers. At high fluence, the heat-affected zone only exists at the edge position. At low flux, more crystallographic defects were found at the edge of the heat-affected zone. In contrast, the laser-induced thermal crack propagation is considered to be a promising process for silicon cutting [28,29,30]. This is because it has the advantage of avoiding these problems [31,32,33] and producing a good surface finish [34,35].

However, there is limited research on the impact of laser processing parameters on cutting quality and optimizing relevant parameters can effectively improve the wafer cutting quality. Therefore, this paper is aimed at the low ablation area of laser wafer cutting and the urgent need for high-efficiency wafer cutting to carry out the research of laser wafer cutting technology. A laser cutting experiment of single crystal silicon was designed with the basic requirement of reducing the laser cutting times and the main objective of optimizing the parameters of laser scanning speed, frequency, and pulse width.

## 2. Experiment Setup

### 2.1. Materials and Methods

#### 2.1.1. Wafer Material

The material used in this experiment was single-sided polished silicon wafer which is a kind of brittle and hard material. After the silicon crystal was crystallized and grown by the straight pulling method, one of the surfaces was in a mirror and shiny state after mechanical or chemical treatment. Monocrystalline silicon is a kind of crystal with a basically complete lattice structure. It has different properties in different crystal directions and is a good semi-conductive material. The surface of a complete silicon chip is like a mirror and the overall shape is like a circle with a small part cut off on the perimeter, hence it is also called a wafer. The direction of the tangent line is the direction of the 0° crystal direction. Table 1 shows the main parameters of the chip used in the experiment.

#### 2.1.2. Wafer-Cutting Morphology and the Judgment Method

This experiment aimed to explore the influence of laser cutting on the surface quality and thermal damage of wafer materials in order to obtain higher quality and less damaged wafers. Similarly to traditional cutting, laser cutting makes use of the high-temperature characteristics of laser cutting to rapidly melt the material at the cutting position and sputter outward, leaving a fine line on the surface of the material. With the increase in cutting times, the fine line gradually moved to the inside of the material, just like the feeding in the turning process, until the bottom of the material is cut off. At the end of each cut, the focus of the laser beam was repositioned to the bottom of the material to begin the next cut. Figure 1 gives the measured view for the wafer cutting.

#### 2.1.3. Laser Processing Equipment

This experiment used a Huaray Polar-355 nanosecond laser for the wafer-cutting experiment, the detailed parameters are given in Table 2. The laser is a pulse laser which can increase and emit laser energy in a very short time as well as through a high frequency of light to complete the material cutting. Due to the light weight of the silicon wafer and the thermal deformation phenomenon during the cutting process, the cutting positions of the silicon wafer vary with different cutting times. Therefore, during the experiment, it was necessary to fix the material to ensure that each cut was carried out on one groove.

#### 2.1.4. Selection of the Measuring Instrument

In this experiment, the Hirox KH-7700 digital 3D video microscope was selected as the parameter-measuring instrument. The instrument uses optical zoom to observe the 2D image of the selected altitude layer and can obtain the 3D vertical image of the selected point through multi-layer superposition technology and Z-axis autonomous movement so as to measure the relevant parameters required for the experiment. The measuring instrument is shown in Figure 2.

### 2.2. Laser Cutting Process Parameters Selection

The materials used in this study included single-sided polished silicon wafers which are prone to heat damage in the process of laser cutting. Therefore, this experiment mainly studied the influence of different laser process parameters on the surface quality and thermal damage size of the wafer after laser cutting and finally completed the optimization of laser parameters. Considering the requirements of mass production for material quality, the production efficiency in the final application, and the linear correlation between production time and cutting times, with the increase in cutting times the surface machining affected zone also increased, so the cutting times were taken as the primary research object to find the minimum cutting times that can cut the material exactly.

In this experiment, the first experiment was the cutting times as the independent variable and other parameters such as laser frequency and scanning speed remained unchanged. Laser parameters: scanning speed: 200 mm/s, laser frequency: 40 kHz, pulse width: 0.3 ns, cutting times: 5, 10, 15, 20, 30, …, 200, 300 times. The minimum cutting times were obtained when the wafer was cut off.

Due to the previously set experimental parameters ranging from 5 to 300 cuts, with a large parameter span, in order to ensure the accuracy of the experimental results, after determining the optimal cutting times (X) that meet the cutting requirements, based on X, 11 sets of laser cutting experiments were added to further determine the minimum cutting times. 

After finding the minimum cutting times that meet the experimental requirements, a control experiment was designed around the cutting times to explore the impact of different laser processes on the damage of cutting quality. The specific experimental parameter design is shown in Table 3.

### 2.3. Overall Experiments Design

The process of laser cutting wafer is essentially a process of drilling which belongs to the vaporization cutting. Lasers have high energy and power. Taking advantage of these characteristics, laser energy accumulates massively on the wafer surface, the material rapidly heats up and vaporizes, some materials are blown away as slag by auxiliary gas, the relative movement of laser and wafer forms a groove, and the laser generates energy accumulation and heat conduction on the wafer surface, leading to the formation of machining affected zones on both sides of the groove. Figure 3 illustrates the development of Wafer Cutting.

As the surface parameters can be obtained directly from laser cutting experiments, the width of the groove and the width of the machining affected zone directly reflect the quality of laser cutting. By sorting out and analyzing the material surface parameters obtained from the previous laser cutting experiment, we can obtain the influence of the cutting times and laser process parameters on the wafer cutting.

During the experiment, by wiping the material surface after cutting, it was found that the machining affected zone on the material surface and the recast layer near the surface on both sides of the groove could be directly removed. In the actual production, if you can use this phenomenon you can increase the utilization rate of materials and reduce production costs, hence the width of the cutting seam and the width of the morphology after the removal of chips are the research objects in order to explore the influence of different laser parameters on its rule. The experimental process is shown in Figure 4 and Figure 5.

This paper focused on the quality control and damage analysis of laser cutting of monocrystalline silicon. In this experiment, firstly, an experimental platform should be established and the appropriate instruments selected. Then, the monocrystalline silicon wafer should be pretreated. Through the laser cutting wafer experiment, the materials should be cut with different cutting times and laser parameters in order to analyze the morphology of the material surface and the machining affected zone. Finally, the laser-cutting mechanism of monocrystalline silicon should be formed.

## 3. Analysis of the Wafer Cutting Mechanism

### 3.1. Wafer Cutting Process Analysis

#### 3.1.1. Judgment of the Threshold Value of Laser Cutting Times

According to the previous experimental design, the initial experiment was carried out with the cutting times as the independent variable and the laser frequency, speed, and other parameters remained unchanged. The parameter values not given in the table are the scanning speed: 200 mm/s; laser frequency: 40 kHz; and pulse width: 0.3 ns. After the laser cutting was completed, the material surface was observed with a three-dimensional microscope and then an alcoholic cotton ball was used to wipe both sides of the groove to remove the chips; after observation of the chips this process was repeated if necessary. The data obtained were sorted out.

The experiment results show that the materials were cut off with the increase in cutting times from the 20th time. Hence, this experiment shows that the cutting times of about 20 times are the lowest times of material cutting. The experiment contents of the laser cutting experiment are shown in Table 4 below.

#### 3.1.2. Determination of Laser Cutting Times

Based on the results of the first experiment for 20 times, we added 11 groups of laser cutting experiments and identified the burn conditions. The identification of the thermal damage and cutting conditions of six groups of laser cutting experiments in which the cutting times were even shows that with the increase in the test times, the material’s groove width, the width of the machining affected zone, and the groove width after chip removal have an increasing trend. After observing the crack after chip removal, it was found that both sides of the original groove with the recast layer were more uniform and closer to a straight line, while both sides of the groove after chip removal were uneven and closer to a broken line. Moreover, the range of broken lines corresponding to different laser parameters was different, so the groove morphology after chip removal was taken as another research object for experimental reference. The specific experimental parameters and cutoff conditions are shown in Table 5.

#### 3.1.3. Analysis of Laser Cutting Quality

Analysis of the above data shows that with the increase in cutting times, the cutting depth also increases, and when the cutting times is 20 times, the material is completely cut off. This is because cutting laser energy is limited: a single cut cannot directly penetrate through the material. If lasers irradiate the surface of the material every time, it will feed a distance into the material until cut off. The following figures show the material morphology under different cutting times. Figure 6 shows the morphology of the material before chip removal under different cutting times.

### 3.2. Micromorphology and Machining Affected Zone Analysis

#### 3.2.1. Material Crack Width Analysis

With the change in laser cutting times, the crack width on the material surface will change. Figure 7 shows the changing trend of the crack width on the material surface with the change in cutting times. Through analysis, the influence rule of laser cutting times on the crack width was obtained: when the laser cutting times are low (less than 80 times), the width of the groove increases with the increase in the cutting times, but when the laser cutting times exceed 80 times, the groove width will rapidly decrease and again slowly increase. This is because the process of laser cutting is a process of melting, decomposition, and the vaporization of materials as well as the spraying and solidification of molten materials. Finally, a certain thickness of the recast layer will be formed on both sides of the cutting seam. When the cutting times are low, the width of the cutting seam is less affected by the thickness of the recast layer so it gradually increases with the cutting times. When the cutting times reach a certain value, at this time, the groove width of laser cutting is more affected by the width of the recast layer than by the direct impact of laser cutting so the groove width decreases with the increase in the laser cutting times. If the cutting times continue to increase, the temperature rises, the recast layer material vaporizes, the thickness becomes smaller, and then the groove width increases again. Figure 8 shows the change in surface crack width of the material as the cutting times increase.

#### 3.2.2. Material Machining Affected Zone Analysis

With the change in laser cutting times, the width of the machining affected zone will also change. Figure 9 shows the trend of the width of the machining affected zone on the material surface changing with the cutting times. Through analysis, the influence rule of laser cutting times on the crack width is obtained: with the increase in laser cutting times, the width of the machining affected zone of materials gradually increases and finally tends to be stable. This is because when the laser cutting times are low, the surface temperature of the material increases with the increase in the cutting times, and gradually extends along the material surface to both sides of the cutting seam. When the cutting times reach a certain value, even if the temperature at the material cutting position continues to increase with the cutting times, the machining affected zone will not continue to extend to both sides due to the thermal conductivity of the material itself, so the width of the machining affected zone at the edge position tends to be stable.

#### 3.2.3. Analysis of Crack Width and Morphology after Chip Removal

Figure 10 shows the crack width of the material after chip removal on the material surface changing with the cutting times. By analyzing the surface morphology of the material after chip removal, the following conclusions were obtained. (1) Crack width after chip removal: the crack width after chip removal increases with the increase in cutting times, which is different from the rule that the crack width without chip removal is affected by the change in cutting times. This is because the use of alcohol cotton balls not only erased the machining affected zone on both sides of the groove but also erased the recast layer near the material surface at the material notch. Therefore, the groove width at this time is only affected by the laser energy received during the experiment and is positively related to it. (2) Width of groove morphology after chip removal: The change in cutting times will have a certain impact on the width of groove morphology after chip removal but the rule is not significant enough. Mainly because there are fewer cutting times and the impact of chips is relatively small. When the cutting times are 20, the crack width is small.

## 4. Laser Experiment and Analysis

### 4.1. Process Analysis of the Influence of Laser Cutting Speed

#### 4.1.1. Process Parameter Design

According to the previous experimental results, 20 times of laser cutting is the best experimental parameter. Keep other process parameters unchanged and the cutting times is 20. Carry out Scientific control with the laser speed as an independent variable. Use a three-dimensional microscope to analyze and observe the cut material surface, and process and summarize the data. The specific experimental results are shown in Table 6.

#### 4.1.2. Morphology Analysis before Chip Removal

By adjusting the laser scanning speed on the crack width and machining, the affected zone width was analyzed and the following conclusion was obtained. (1) Cutting effect: the cutting times under the condition of fixed for 20 times, the lower the laser scanning speed is, the more obvious the cutting effect is. This is because the laser is not continuous but has pulse properties. Keeping the cutting times unchanged, the slower the scanning speed is, the more times the material is scanned per unit length, and the greater the cutting depth is. (2) Crack width: when the laser scanning speed exceeds 100 mm/s, the groove width gradually decreases with the increase in the laser scanning speed but when the laser scanning speed is too small (50 mm/s), the groove width also has a decreasing trend. This is because when the laser frequency is unchanged, the number of laser pulses emitted per unit time is the same, the laser scanning speed increases, the number of pulses per unit length of the material decreases, so the groove width becomes smaller. However, when the scanning speed is too small, the material temperature increases, the thickness of recast layer increases, and the groove width becomes smaller. (3) Width of machining affected zone: with the increase in scanning speed, the width of the material’s machining affected zone gradually decreases. The reason is the same as the analysis of the effect of the laser scanning speed on the groove width. Figure 11 shows the changing trend of the relevant area with the scanning speed and Figure 12 shows the material morphology at a partial scanning speed.

#### 4.1.3. Morphology Analysis after Chip Removal

By analyzing the width of the chip morphology and groove morphology at different laser scanning speeds, the following conclusions are obtained. (1) Crack width after chip removal: the crack width after chip removal decreases with the increase in scanning speed because the alcohol cotton ball not only erases the machining affected zone on both sides of the groove but also erases the recast layer at the material notch; at this time, the groove width is only affected by the laser energy during the experiment and is positively correlated with it. (2) Groove profile width: the groove profile width increases with the increase in scanning speed and decreases with the decrease in scanning speed. Figure 13 shows the changing trend of the relevant area with the laser scanning speed. Figure 14 shows the (a) groove width and (b) groove morphology after cutting at a partial laser speed.

### 4.2. Process Analysis of the Influence of Laser Frequency

#### 4.2.1. Process Parameter Design

According to the previous experimental results, the number of cutting times was determined to be 20, and other process parameters remained unchanged. The Scientific control was carried out with the laser frequency as the independent variable. In the process of the experiment, the ordinary laser mean power is equal to the product of the pulse energy and frequency. The formula for the laser used in this experiment can be used with the corresponding laser mean power frequency. Table 7 shows their corresponding relationships.

Therefore, when laser frequency is involved in the table, the unit name used is laser power/W (frequency/kHz). A three-dimensional microscope was used to observe the cut material surface and sort out the data obtained, as shown in Table 8.

#### 4.2.2. Morphology Analysis before Chip Removal

Through the analysis of the crack width and machining affected zone width under different laser frequencies, the following conclusions can be obtained. (1) Cutting effect: when the cutting times are 20 times, the higher the laser frequency, the more obvious the cutting effect. This is because when the cutting time is unchanged, the higher the laser frequency, the more times the material is pulsed by the laser per unit length, and the greater the cutting depth. (2) Crack width: with the increase in laser frequency, the groove width of the material gradually increases. This is because of the average power of the laser = laser frequency * single pulse energy. When the single pulse energy of the laser remains unchanged, the frequency is increased. If the energy received by the material in unit time increases, the groove width increases. (3) Width of the machining affected zone: with the increase in laser frequency, the width of the machining affected zone of the material shows a trend of decreasing gradually. The peak intensity at the center of the spot is inversely proportional to the laser pulse frequency. The calculation method of the laser pulse energy (*E_p_*) is the following:(1)$${\;E}_{p}=\frac{\pi }{8}\tau_{p}d_{p}^{2}I_{p}$$

Therefore, as the laser frequency increases, the peak intensity (*I_p_*) at the center of the spot decreases. At this time, the laser pulse energy (*E_p_*) decreases, the heating ability of the laser weakens, and the liquid molten material ejected during the laser cutting process decreases. So even if the increase in laser frequency will increase the number of lasers of material per unit length, this effect does not surpass caused by laser heating ability reduce spewed molten material to reduce the effects of the liquid, so with the increase in laser frequency, but the machining affected zone width decreases. Figure 15 shows the changing trend of the relevant region with the laser frequency. Figure 16 Material morphology at partial laser frequencies (a) material morphology before chip removal (b) amplification at the groove.

#### 4.2.3. Morphology Analysis after Chip Removal

Through the analysis of the morphology width and groove morphology width after chip removal at different laser speeds, the following conclusions were obtained under the current experimental conditions. (1) Crack width after chip removal: For the effect of laser frequency on the crack width after chip removal, the crack width after chip removal increases with the increase in power; this is because with the laser parameters according to the trend change, the laser energy absorbed on the unit length of the groove increases, the local temperature increases, the material continues to melt, so the groove increases. (2) Groove morphology width: groove morphology width decreases with the increase in laser frequency and power and vice versa. Figure 17 shows the changing trend of the relevant region with the laser velocity and Figure 18 shows (a) the width of the groove and (b) the morphology of the groove after the chip removal at part of the laser frequency.

### 4.3. Process Analysis of the Influence of Laser Pulse Width

#### 4.3.1. Process Parameter Design

According to the previous experimental design, the best cutting number of 20 times is achieved when other laser processes are unchanged, namely with the laser pulse width as the independent variable control experiment, by using a three-dimensional microscope to look at the material surface after cutting to obtain the material surface crack width under different pulse width process and by machining affected zone width data., A summary of the specific experimental parameters after data processing are shown in Table 9.

#### 4.3.2. Morphology Analysis before Chip Removal

By analyzing the crack width and machining affected zone width under different pulse widths, the following conclusions are obtained. (1) Cutting effect: during the pulse width experiment, the cutting times are fixed at 20 times and the materials are cut off. Changing the pulse width does not have a significant impact on this result, so the pulse width is not a laser parameter that has a significant impact on the cutting depth. (2) The pulse width of laser cutting has a certain influence on the groove width but it does not have regularity. It is certain that the groove is narrower when the pulse width is 0.3 ns, that is, the cutting quality is better. (3) The same as the influence of pulse width on the groove width, the pulse width of laser cutting has a certain influence on the width of the machining affected zone but it has no regularity. When the pulse width is 0.3 ns, the width of the machining affected zone is smaller so the laser cutting quality is better. Figure 19 shows the changing trend of relevant areas with pulse width. Figure 20 shoes the material morphology under a partial pulse width: (a) material morphology before chip removal and (b) amplification at the groove.

#### 4.3.3. Morphology Analysis after Chip Removal

Through the analysis of the width of chip morphology and the width of groove morphology after chip removal with different pulse widths, the following conclusions are obtained. (1) Crack width after chip removal: The crack width after chip removal increases with the increase in pulse width because when the laser parameters change according to the trend, the laser energy absorbed on the unit length of the groove increases, the local temperature rises, and the material continues to melt, so the groove increases. (2) Groove morphology width: the surface morphology of the groove can be affected by changing the laser pulse width but there is no regularity. When the laser pulse width is 0.3 ns, the surface morphology width of the groove is small, that is, the cutting quality is good. Figure 21 shows the changing trend of relevant areas with pulse width, Figure 22. Material morphology under partial pulse width after chip removal (a) The width of the groove (b) amplification at the groove.

## 5. Conclusions

In view of the urgent demand of the current market for efficient wafer scribing, this paper has carried out research on the quality control and damage analysis of laser cutting of monocrystalline silicon. Through laser cutting experiments and the treatment of the material surface after cutting, the existence of thermal damage caused by laser cutting of monocrystalline silicon under the current experimental conditions and the influence of different laser parameters on the material surface quality after cutting were obtained.

(1) When laser cutting monocrystalline silicon, in addition to forming a groove at the cutting position a machining affected zone will also be formed on both sides of the groove. The size of the two zones will directly affect the cutting quality. After the material surface is wiped, the machining affected zone and the recast layer near the upper surface in the groove can be directly removed. The surface quality after wiping is basically the same as that before cutting. In practical production, eliminating the machining affected zone by this method can improve the utilization rate of materials.

(2) With the increase in laser cutting times, the width of the material’s machining affected zone and the crack width after chip removal also increase. The original crack width will also increase with the increase in cutting times when the cutting times are low but it will rapidly decrease when the cutting times exceed 80 and then slowly increase. At the same time, the change in cutting times will affect the surface morphology of the groove and the surface morphology is better when the cutting times are 20.

(3) With the decrease in the laser scanning speed, the width of the material’s groove morphology decreases but the width of the material’s machining affected zone and the crack width after chip removal show an increasing trend. The original crack width also shows an increasing trend when the cutting speed decreases but it will decrease rapidly when the speed decreases to 50 mm/s. The increase in laser frequency will increase the crack width and the crack width after chip removal but decrease the crack morphology and the width of the machining affected zone. The laser pulse width has a certain effect on the cutting quality but the regularity is not obvious. When the pulse width is between 0.3 ns and 0.45, the cutting quality is better, and while when the pulse width is 0.15 ns, the cutting quality is the worst.

## Figures and Tables

**Figure 1 micromachines-14-01542-f001:**
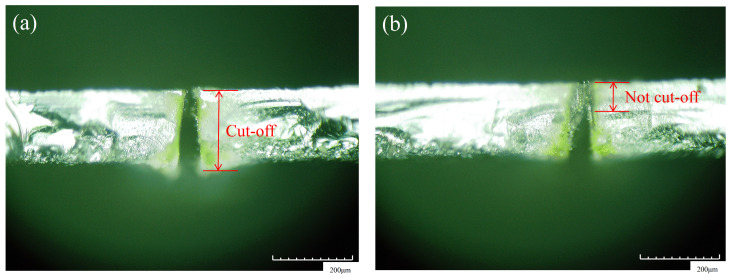
Schematic diagram of whether the material is cut-off or not. (**a**) Cut-off and (**b**) not cut-off.

**Figure 2 micromachines-14-01542-f002:**
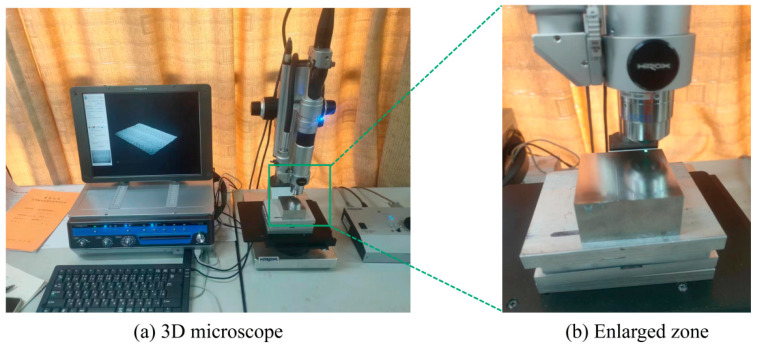
Measuring instrument.

**Figure 3 micromachines-14-01542-f003:**
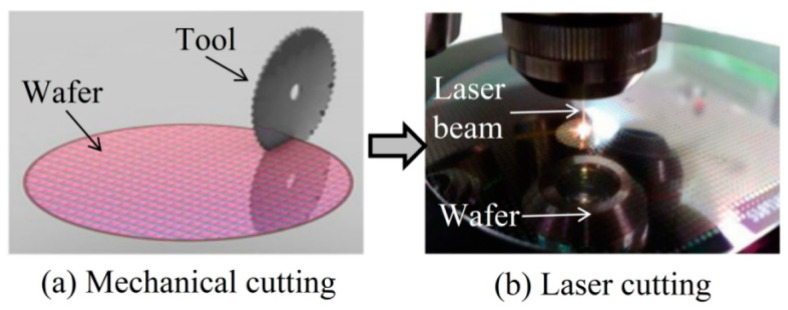
Development of Wafer Cutting.

**Figure 4 micromachines-14-01542-f004:**
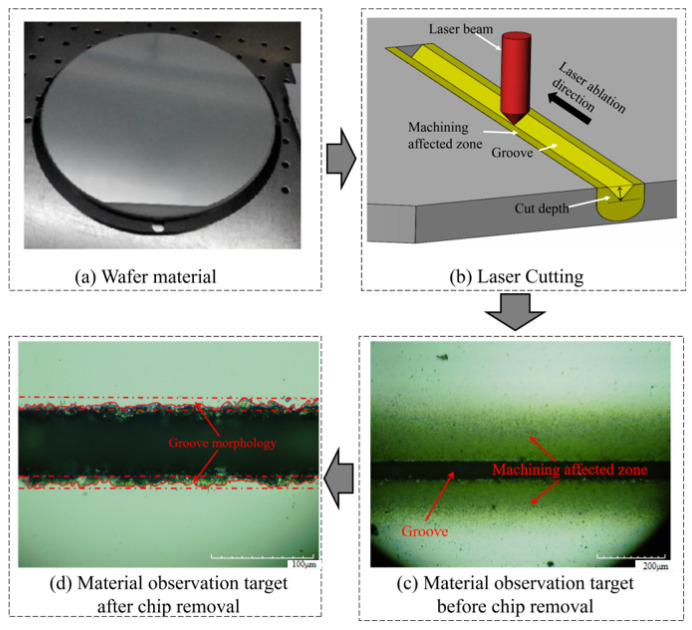
Laser cutting and the observation target.

**Figure 5 micromachines-14-01542-f005:**
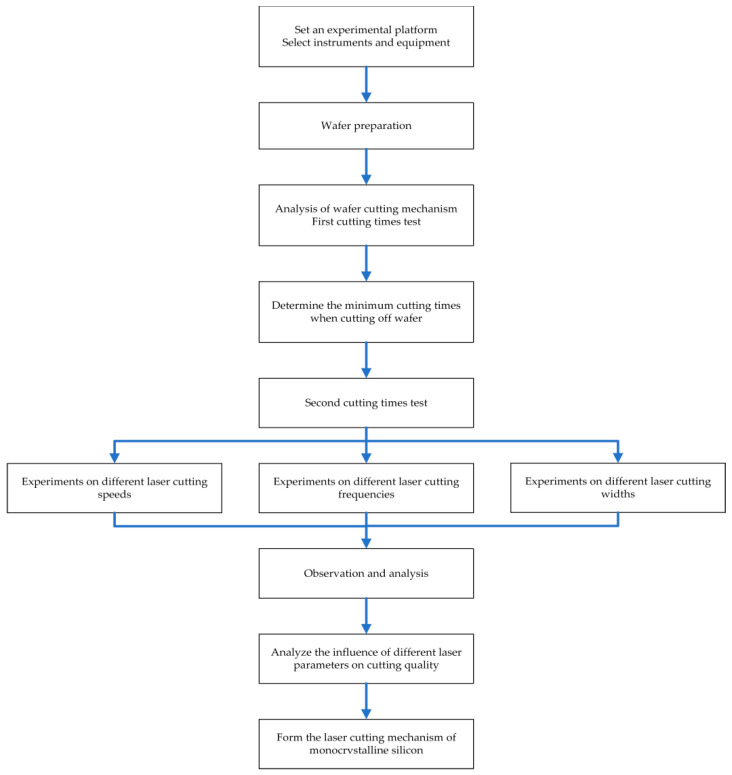
Schematic Diagram.

**Figure 6 micromachines-14-01542-f006:**
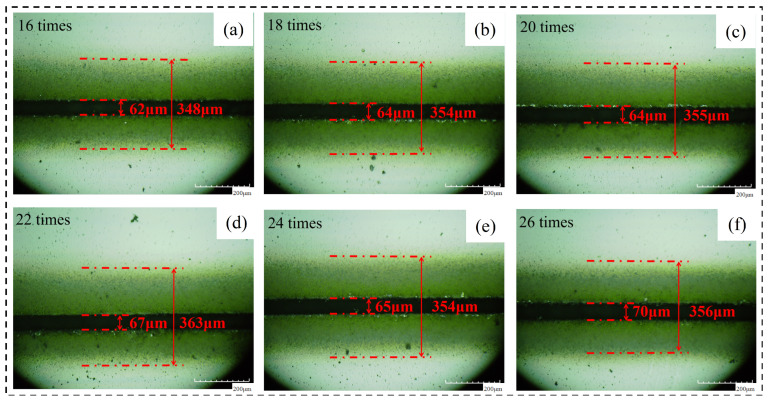
The material morphology before chip removal. (**a**) 16 times, (**b**) 18 times, (**c**) 20 times, (**d**) 22 times, (**e**) 24 times, (**f**) 26 times.

**Figure 7 micromachines-14-01542-f007:**
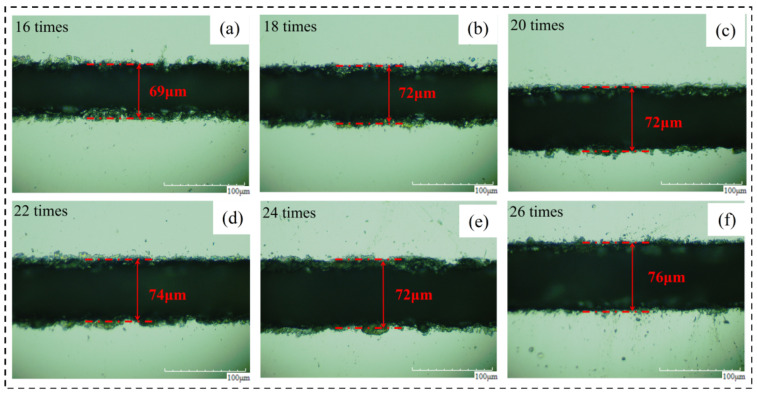
The material morphology after chip removal. (**a**) 16 times, (**b**) 18 times, (**c**) 20 times, (**d**) 22 times, (**e**) 24 times, (**f**) 26 times.

**Figure 8 micromachines-14-01542-f008:**
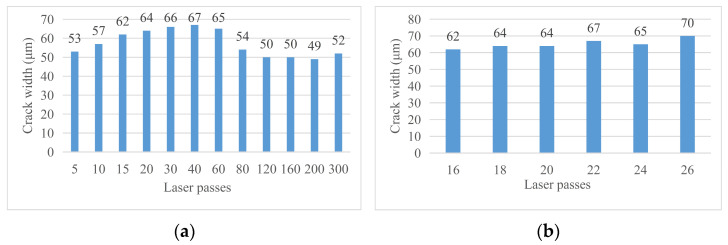
Curve of the surface crack width changing with cutting times. (**a**) Crack width after 5–300 times of cutting and (**b**) crack width after 16–26 times of cutting.

**Figure 9 micromachines-14-01542-f009:**
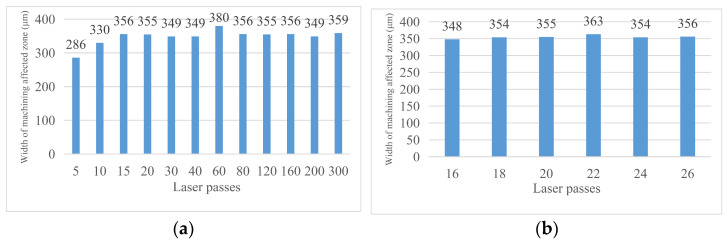
Curve of the width of the machining affected zone changes with cutting times. (**a**) Cutting 5–300 times and (**b**) cutting 16–26 times.

**Figure 10 micromachines-14-01542-f010:**
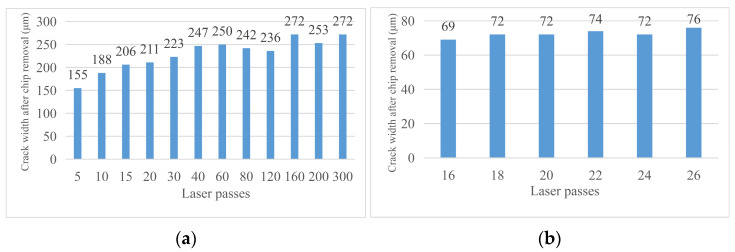
Crack width of the material after chip removal. (**a**) Cutting 5–300 times and (**b**) cutting 16–26 times.

**Figure 11 micromachines-14-01542-f011:**
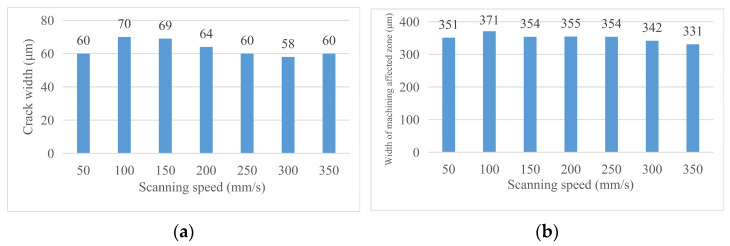
Change in material morphology before chip removal in speed experiment. (**a**) Material crack width. (**b**) material machining affected zone width.

**Figure 12 micromachines-14-01542-f012:**
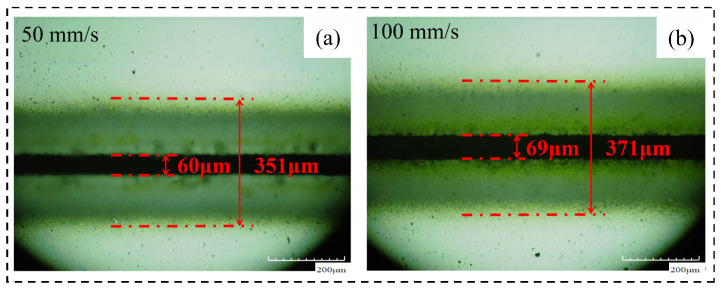
Material morphology at a partial cutting speed. (**a**) 50 mm/s, (**b**) 100 mm/s.

**Figure 13 micromachines-14-01542-f013:**
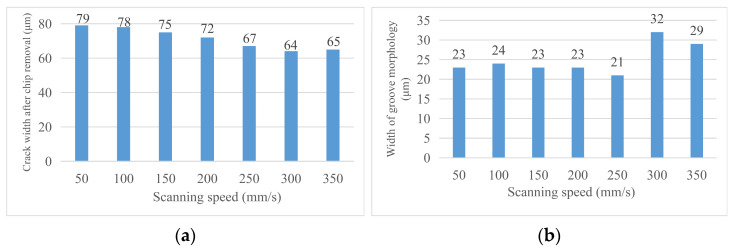
Change in material morphology after chip removal in speed experiment. (**a**) Groove width after chip removal and (**b**) groove morphology width.

**Figure 14 micromachines-14-01542-f014:**
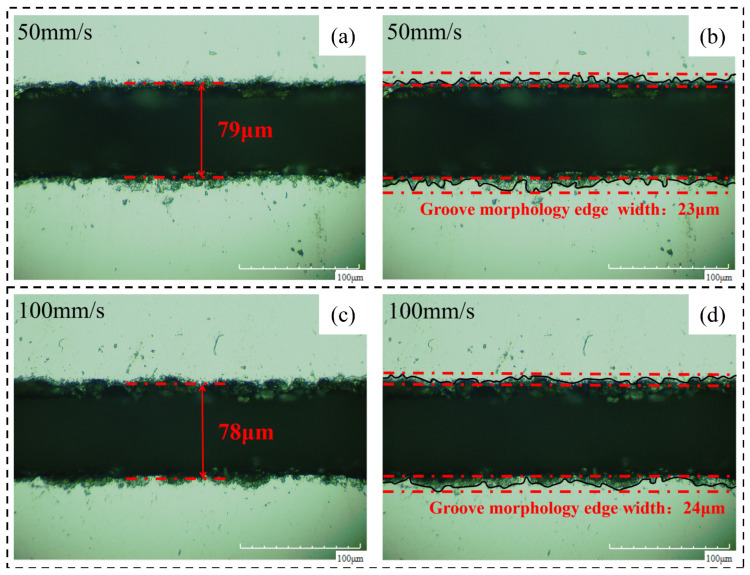
Material morphology after chip removal at partial cutting speed. (**a**) 50 mm/s groove width, (**b**) 50 mm/s groove morphology edge width, (**c**) 100 mm/s groove width, (**d**) 100 mm/s groove morphology edge width.

**Figure 15 micromachines-14-01542-f015:**
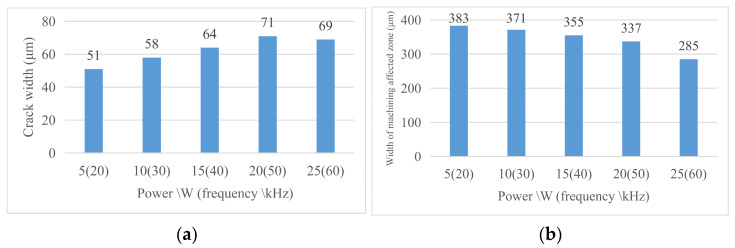
Change in material morphology before chip removal in frequency experiment. (**a**) Material crack width and (**b**) material machining affected zone width.

**Figure 16 micromachines-14-01542-f016:**
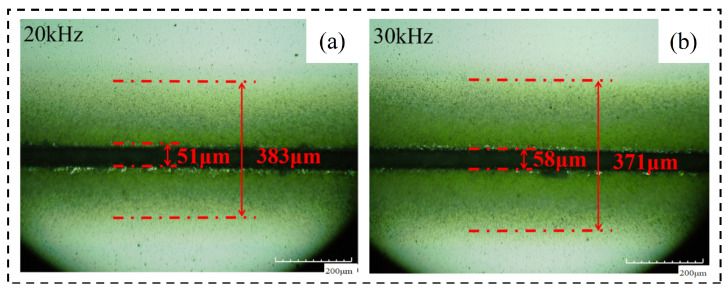
Material morphology at partial laser frequencies. (**a**) 20 kHz, (**b**) 30 kHz.

**Figure 17 micromachines-14-01542-f017:**
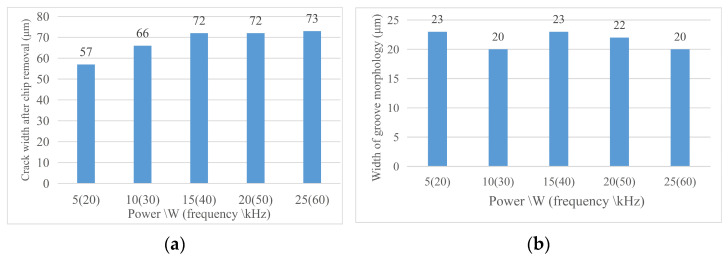
Change in material morphology after chip removal in frequency experiment. (**a**) Groove width after chip removal and (**b**) groove morphology width.

**Figure 18 micromachines-14-01542-f018:**
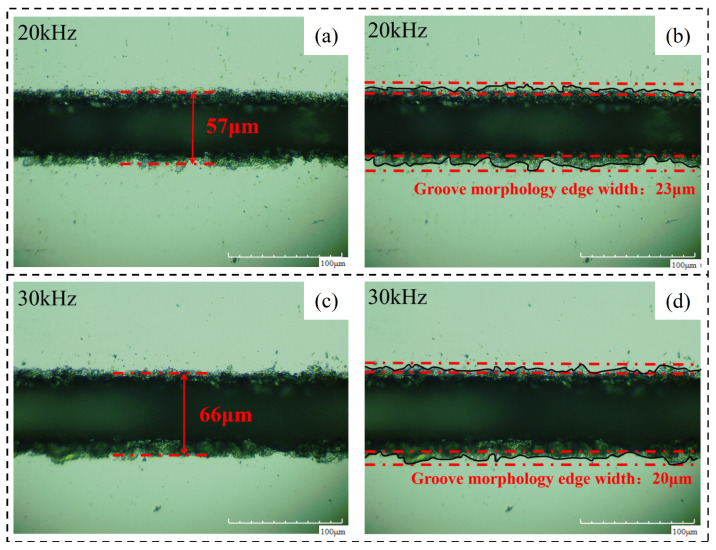
Material morphology after chip removal at partial laser frequencies. (**a**) 20 kHz groove width, (**b**) 20 kHz groove morphology edge width, (**c**) 30 kHz groove width, (**d**) 30 kHz groove morphology edge width.

**Figure 19 micromachines-14-01542-f019:**
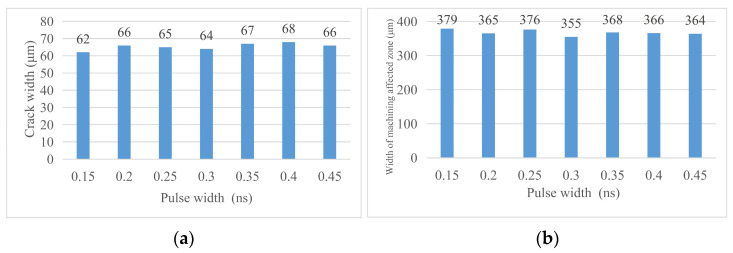
Change in material morphology before chip removal in the pulse width experiment. (**a**) Material crack width and (**b**) material machining the affected zone width.

**Figure 20 micromachines-14-01542-f020:**
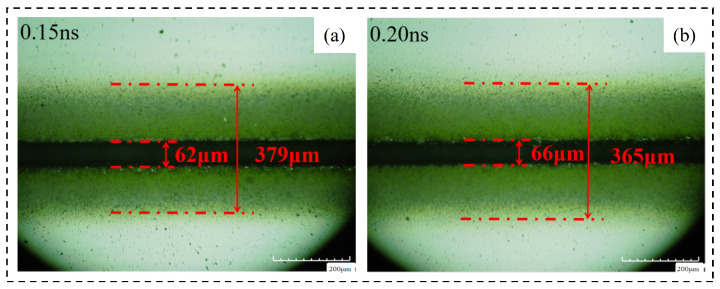
Material morphology at partial laser pulse width. (**a**) 0.15 ns, (**b**) 0.20 ns.

**Figure 21 micromachines-14-01542-f021:**
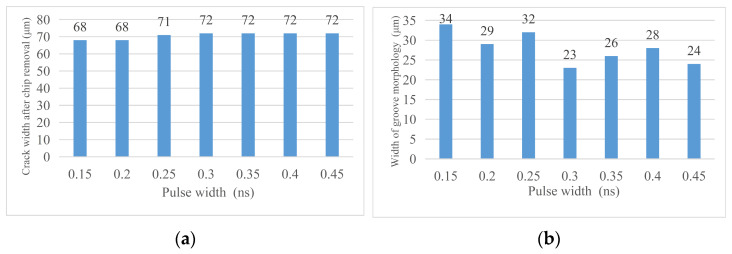
Change in material morphology after chip removal in the pulse width experiment. (**a**) Groove width after chip removal and (**b**) groove morphology width.

**Figure 22 micromachines-14-01542-f022:**
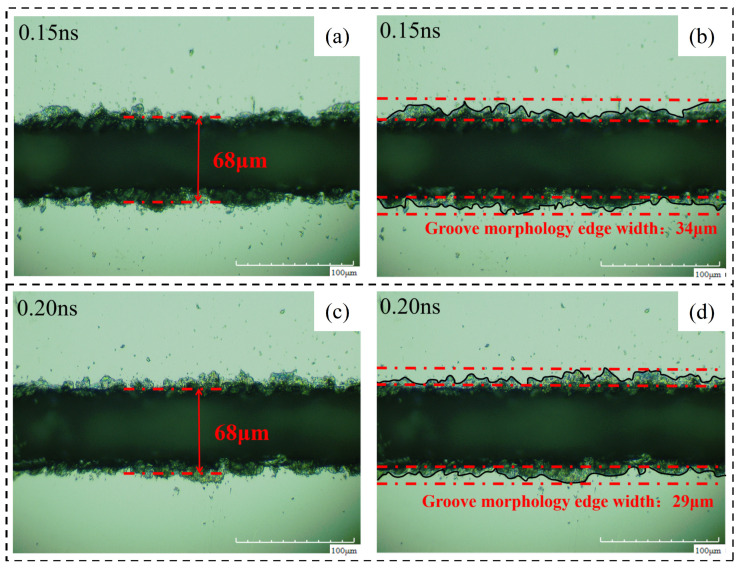
Material morphology after chip removal at a partial laser pulse width. (**a**) 0.15 ns groove width, (**b**) 0.15 ns groove morphology edge width, (**c**) 0.20 ns groove width, (**d**) 0.20 ns groove morphology edge width.

**Table 1 micromachines-14-01542-t001:** Wafer Parameters.

Basic Parameter			
**Type**	Single side polished silicon wafer	**Thickness**	200 ± 10 μm
**Model/crystal orientation**	P<100>	**Diameter**	100 ± 0.4 mm
**Growth method**	CZ	**Resistivity**	1–10 Ωcm

**Table 2 micromachines-14-01542-t002:** Parameters of the Huaray Polar-355 nanosecond laser.

Basic Parameters			
Output power	>5 W@50 kHz	Working temperature	10–35 °C
Single pulse energy	>125 μJ@40 kHz	Cooling method	Water
Pulse width	16 ± 2 ns@50 kHz	Voltage	110/220 V, 50/60 Hz
Start time	<15 min	Focus (*z*-axis direction)	33 cm
Repetition frequency	20 kHz–200 kHz	Spatial mode	TEM_00_ (M^2^ ≤ 1.2)
Power stability	≤3% rms	Pulse stability	≤3% rms
Beam diameter	<8 m	Spot divergence angle	≤1 mrad
Beam directivity	<25 μrad	Rad spot roundness	≥90%
Manufacturer	Wuhan Huaray Precision Laser Co., Ltd.	Nation	Wuhan, China

**Table 3 micromachines-14-01542-t003:** Experimental test and analysis.

	Parameters
**Scanning speed/mm·s^−1^**	50, 100, 150, 200, 250, 300, 350
**Laser frequency/kHz**	20, 30, 40, 50, 60
**Pulse width/ns**	0.15, 0.2, 0.25, 0.3, 0.35, 0.4, 0.45
**Laser passes**	20

**Table 4 micromachines-14-01542-t004:** First experiment parameters are based on the change in cutting times.

Laser Passes	Crack Width/μm	Width of Machining Affected Zone/μm	Crack Width after Chip Removal/μm	Whether It Is Cut Off
5	53	286	155	No
10	57	330	188	No
15	62	356	206	No
20	64	355	211	Yes
30	66	349	223	Yes
40	67	349	247	Yes
60	65	380	250	Yes
80	54	356	242	Yes
120	50	355	236	Yes
160	50	356	272	Yes
200	49	349	253	Yes
300	52	359	272	Yes

**Table 5 micromachines-14-01542-t005:** Parameters of the second experiment based on the change in cutting times.

Laser Passes	Crack Width/μm	Width of Machining Affected Zone/μm	Crack Width after Chip Removal/μm	Width of Groove Morphology/μm	Whether It Is Cut Off
16	62	348	69	24	No
18	64	354	72	24	No
20	64	355	72	23	Yes
22	67	363	74	27	Yes
24	65	354	72	28	Yes
26	70	356	76	20	Yes

**Table 6 micromachines-14-01542-t006:** Experimental data based on the scanning speed.

Scanning Speed/mm·s^−1^	Crack Width/μm	Width of Machining Affected Zone/μm	Crack Width after Chip Removal/μm	Width of Groove Morphology/μm	Whether It Is Cut Off
50	60	351	79	23	Yes
100	70	371	78	24	Yes
150	69	354	75	23	Yes
200	64	355	72	23	Yes
250	60	354	67	21	Yes
300	58	342	64	32	No
350	60	331	65	29	No

**Table 7 micromachines-14-01542-t007:** Laser frequency and power.

**Frequency/kHz**	20	30	40	50	60
**Average power/W**	5	10	15	20	25

**Table 8 micromachines-14-01542-t008:** Experimental data based on laser frequency.

Power/W (Frequency/kHz)	Crack Width/μm	Width of Machining Affected Zone/μm	Crack Width after Chip Removal/μm	Width of Groove Morphology/μm	Cut Off
5 (20)	51	383	57	23	No
10 (30)	58	371	66	20	Yes
15 (40)	64	355	72	23	Yes
20 (50)	71	337	72	22	Yes
25 (60)	69	285	73	20	Yes

**Table 9 micromachines-14-01542-t009:** Experimental parameters based on the pulse width.

	Crack Width/μm	Width of Machining Affected Zone/μm	Crack Width after Chip Removal/μm	Width of Groove Morphology/μm	Whether It Is Cut Off
0.15	62	379	68	34	Yes
0.2	66	365	68	29	Yes
0.25	65	376	71	32	Yes
0.3	64	355	72	23	Yes
0.35	67	368	72	26	Yes
0.4	68	366	72	28	Yes
0.45	66	364	72	24	Yes

## Data Availability

Not applicable.

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
