# Peer review of "Process Analysis and Topography Evaluation for Monocrystalline Silicon Laser Cutting-Off"

_micromachines, 2023, doi:10.3390/mi14081542_

Round 1

Reviewer 1 Report

1. The abstract is suggested to be revised with more direct conclusion with specific data.

2. It is recommended that the bar chart of the trend change in this paper be supplemented with a line trend, and that the colour of the chart be made more vivid so that the trend change can be clearly indicated.

3. In 3.2.3 of this paper, it is suggested to explain why the width of the groove profile does not change significantly after chip removal be explained.

4. The scale should be clearly shown in each of the post-processing material profiles.

5. Some of the references are too old and it is recommended that more recent articles be cited.

6. The schematic flow diagram in Figure 5 can be reduced and a concise introduction is sufficient.

Author Response

The Editorial Office

Micromachines

June 25th, 2023

Dear Editor in Chief:

     Thank you very much to give us this opportunity to revise our manuscript entitled Process analysis and topography evaluation for monocrystalline silicon laser cutting-off (Manuscript ID: Micromachines-2460996) for Micromachines. We appreciate your contribution for selecting the appropriate reviewers for our manuscript. We really do value the comments and suggestions that let our work be more clarified.

     Please find attached (below) list of changes in the revised manuscript and our detailed response to each issue raised by the reviewers. The manuscript after considering reviewers’ suggestions has been thoroughly modified.

     Here, we would like to confirm that the revised version of our manuscript has been supported by appropriately supplemented documents. Although some changes in the text have been done according to the suggestions of the reviewers, the exclusive extension in the manuscript is maintained to be adequate with the journal style.

     Please check our amendments in this manuscript. Hopefully, this revised version would merit your consideration.

     Thanks again for your time and contribution for selecting the appropriate reviewers.

Sincerely yours,

Chongjun Wu

June 25th, 2023

Reviewer 2 Report

The objective of this work is to optimize the cutting of monocrystalline silicon chips with a laser emitting in the UV, with a temporal width of fractions of nanoseconds and a repetition rate of kHz. It analyzes the width of the cut as a function of some process parameters and the number of overlapping scans necessary for the laser beam to completely cut the blade.

The text, however, is not clear, contains several conceptual errors, is repetitive, and does not adequately describe the experiments carried out. The reader has great difficulty in understanding how the experiment was done and what the author intended in each experiment.

The description of the materials and methods used is also not complete. There is no description of the focusing system or the part's moving stage. The focusing lens used is not mentioned and therefore it is not possible to calculate the diameter of the focal point and the intensity used in the experiments. This is a crucial point that totally compromises the merit of the work.

The analysis is performed only on the morphology of the cut region, where the width of the cut and a certain affected machining region are analyzed as a function of some process parameters. The analysis is very simple, only phenomenological, and done only through optical microscopy. There is no deeper analysis of the physical and crystalline state of the material in the supposedly heat-affected region. Thus, the scientific merit of the work is greatly impaired.

Other observations are:

The summary looks more like comments on variations of cutting process parameters and their effects on the cutting region. It is not a clear summary of the experiments.

The author talks a lot about machining affected zone“, but does not explain well what this means and what the implications are for the use of the final material. It does not analyze the crystalline structure in the heat-affected zone. There is no scanning electron microscopy image showing the state of the cut surface or the affected region with greater magnification. Microcracks can be produced and affect the physical properties of the material.

“number of laser cutting” – it is also not explained what it means. It is inferred that it is the number of overlapping laser scan lines on the material, but this would need to be well explained. In this case, it is also not said whether the focus of the laser beam is repositioned into the material with each pass of the beam. The part thickness is 200 microns, which is probably greater than the focused beam confocal parameter. This, however, was not mentioned, was not calculated, and cannot be calculated due to the lack of information about the lens used.

The author talks about cleaning the cut region, but does not describe how this is done; only in the middle of the text does he comment on the effect of using a cotton swab with alcohol for this cleaning.

It gives the following information: “light speed is nanosecond level”, an obvious mistake.

Most figures are not named in the text.

Author Response

(The authors gave the same response as above.)

Reviewer 3 Report

The processing technology of monocrystalline silicon is one of the difficult problems to be solved urgently at present. The engineering significance of this paper is feasible. Specifically, this manuscript studies the influence mechanism of different laser processes on the groove size and surrounding area of laser cutting. Overall, the structure of the paper is relatively comprehensive. However, there are still some contents that need to be further improved. The comments are given as follows, which may be helpful to improve the paper quality.

(1) In the introduction, the authors cite a large number of references. However, these references have not been elaborated and briefly discussed. At the same time, it is necessary to increase the detailed elaboration of the contribution of this paper and the difference between it and the existing literature.

(2) Please carefully check whether the schematic diagram (Figure 5) of the method in the paper meets the correct standards.

(3) Suggest adding a comparison of experimental results between mechanical cutting and laser cutting.

(4) In the conclusion, the authors mentioned some technical parameters. It is suggested to provide a reference range, because the actual optimal process parameters are not necessarily a specific parameter, but in a certain range.

The language needs some improvement

Author Response

(The authors gave the same response as above.)

Round 2

Reviewer 2 Report

In this new version, the author makes some modifications without, however, modifying in a profound way what has already been presented previously, and the quality of the English language still needs to improve. It includes a few more references and a few more comments without, however, significantly improving the initial text.

Regarding the experimental arrangement, the author answers some questions posed in the previous analysis, such as cleaning the cut region, repositioning the focus after each cut, without, however, detailing how this is done. The focal length, of the focusing lens that had not been mentioned previously, now appears in a table with all the specifications of the laser system. Here, there was an exaggeration in the presentation of the laser system specifications that have no implications in the presented experiment.

We do not see, in this work, a new contribution to the understanding of the process of cutting silicon wafers. The results are based on phenomena observed in the particular experiment, without an explanation or connection with the physics of the interaction of laser radiation with matter, which makes it impossible to extend the results to other arrangements or materials. The relationship of results is presented as a function of fundamental parameters of the particular system, such as velocity and temporal width. No relationship is shown between the results and the intensity used, the pulse overlapping rate, the fluence and the total energy deposited per unit volume. There is also no analysis of the crystalline structure of the heat-affected region, comparing its structure before and after the experiments.

For these reasons, the conclusion is that the work does not have an adequate level of quality for publication.

Reviewer 3 Report

论文是可以接受的。